# Cerebral Organoid Arrays for Batch Phenotypic Analysis in Sections and Three Dimensions

**DOI:** 10.3390/ijms241813903

**Published:** 2023-09-09

**Authors:** Juan Chen, Haihua Ma, Zhiyu Deng, Qingming Luo, Hui Gong, Ben Long, Xiangning Li

**Affiliations:** 1Britton Chance Center for Biomedical Photonics, Wuhan National Laboratory for Optoelectronics, Huazhong University of Science and Technology, Wuhan 430074, China; 2Key Laboratory of Biomedical Engineering of Hainan Province, School of Biomedical Engineering, Hainan University, Haikou 570228, China; 3HUST-Suzhou Institute for Brainsmatics, Jiangsu Industrial Technology Research Institute, Suzhou 215125, China

**Keywords:** cerebral organoids, array imaging, single-cell phenotypic analysis, aging

## Abstract

Organoids can recapitulate human-specific phenotypes and functions in vivo and have great potential for research in development, disease modeling, and drug screening. Due to the inherent variability among organoids, experiments often require a large sample size. Embedding, staining, and imaging each organoid individually require a lot of reagents and time. Hence, there is an urgent need for fast and efficient methods for analyzing the phenotypic changes in organoids in batches. Here, we provide a comprehensive strategy for array embedding, staining, and imaging of cerebral organoids in both agarose sections and in 3D to analyze the spatial distribution of biomarkers in organoids in situ. We constructed several disease models, particularly an aging model, as examples to demonstrate our strategy for the investigation of the phenotypic analysis of organoids. We fabricated an array mold to produce agarose support with microwells, which hold organoids in place for live/dead imaging. We performed staining and imaging of sectioned organoids embedded in agarose and 3D imaging to examine phenotypic changes in organoids using fluorescence micro-optical sectioning tomography (fMOST) and whole-mount immunostaining. Parallel studies of organoids in arrays using the same staining and imaging parameters enabled easy and reliable comparison among different groups. We were able to track all the data points obtained from every organoid in an embedded array. This strategy could help us study the phenotypic changes in organoids in disease models and drug screening.

## 1. Introduction

Currently, there is an increasing need to understand the underlying mechanisms of aging and aging-related diseases as the world population ages, and cerebral organoids offer an excellent platform to study neurodegenerative diseases such as Alzheimer’s disease [1,2] and Parkinson’s disease [3,4,5]. In addition to using patient-derived cerebral organoids, we could also treat cerebral organoids with certain drugs in vitro to mimic aging or damaged brains in humans. These induced disease models include the D-galactose-induced aging model [6], the hydrogen peroxide-induced oxidative stress model [7], the lipopolysaccharide (LPS)-induced inflammation model [8], the alcohol-induced neurotoxicity model [9,10], etc. These organoid models have contributed to our understanding of disease mechanisms and driven innovation for new therapies.

Imaging technologies are often used to evaluate organoid models. They have aided us in understanding the cellular composition, physiological processes, cytoarchitecture, and cell–cell interactions inside organoids, but there are still challenges that need to be addressed [11]. As organoid culturing techniques developed, some cerebral organoids with high complexity reached several millimeters in diameter, posing a challenge for in situ imaging and phenotypic analysis of organoids [12,13]. As a result, we previously developed a whole-organoid processing pipeline for labeling, embedding, imaging, and analyzing intact millimeter-scale cerebral organoids, which utilized the fMOST system to produce high-resolution 3D datasets, where we also demonstrated the neural rosette structures present in commercial cerebral organoids [14]. In addition to the large size of cerebral organoids, batch processing of organoids is another challenge in the field due to the inherent variability among individuals and batches of cerebral organoids [15]. This variability could be attributed to the fact that each organoid is formed through independent regional development of “neuroepithelial units”, commonly referred to as “neural rosettes”, which makes it difficult to rely on traditional tissue atlas-based 2D histology to locate structures [16]. Traditional methods would require excessively long hours and high costs to process large organoids in bulk. Thus, there is a pressing need for high-throughput methods for the phenotypic analysis of organoids that are suitable for evaluating a variety of disease models.

Several batch processing and imaging methods for spheroids have been developed. Without special accommodation, organoids and spheroids randomly embedded in agarose gels would be on different planes and require manual imaging of each organoid. This obstacle could be overcome by applying the principles of the tissue microarray technique [17]. Researchers embedded spheroids in an array of agarose by first placing them in microwells on an agarose support and then pouring another layer of agarose solution on top [18,19,20,21]. These agarose blocks allowed imaging of multiple spheroids on a single plane at a time after sectioning, reducing the time and effort needed significantly. However, these methods were designed for and only tested on small spheroids with diameters of several hundred micrometers, and our cerebral organoids were larger than one millimeter. Efficient batch processing methods for large cerebral organoids remain a challenge. In addition, imaging of multiple organoids in agarose sections and in 3D can be efficient in situ phenotypic analysis with spatial localization, which is also urgently needed.

Here, we present a strategy to embed, stain, and image organoids in arrays both in agarose sections and in 3D for in situ phenotypic analysis with spatial localization of specific biomarkers. We analyzed brain disease models induced by D-galactose, hydrogen peroxide, and lipopolysaccharides (LPS) by evaluating cell viability and aging hallmarks in treated cerebral organoids. We were also able to track the complete work flow on each organoid in an array for model comparison. This method could become a valuable tool for phenotypic studies in organoid-based disease models and high-throughput drug screening.

## 2. Results

### 2.1. Integrated Strategy for Organoids Phenotypic Analysis in Batch

We developed an integrated strategy for phenotypic analysis of batch organoids in disease modeling and drug screening to meet the need for rapid screening of a large number of organoids. Due to organoids’ inherent variability among individuals and batches, experiments using them usually require a large sample size. First, organoids were cultured and drug-treated in low-adherent 96-well U-bottom plates. These organoids were then transferred to agarose supports for imaging. We conducted three types of imaging: live imaging of whole organoids using live/dead assays in agarose microwells; imaging of agarose sections embedded with organoid arrays; and 3D imaging of whole organoids embedded in agarose blocks using fMOST (Figure 1A). These imaging approaches allowed us to analyze cell viability and phenotypic changes in organoids after drug treatments from both agarose sections and 3D imaging perspectives.

To batch process organoids in arrays, we fabricated array molds for making agarose supports with microwells, which is essential for the arrangement of organoids in arrays. First, we fabricated molds with 3 × 3 or 3 × 5 intrusions using light-sensitive resin and poured boiled agarose solution on top to produce agarose supports with microwells in arrays (Figure 1A). These microwells were hemispherical to prevent the deformation of organoids. Live organoids with live/dead staining were transferred into the microwells of the agarose supports for imaging under a fluorescent stereo zoom microscope to evaluate the distribution of live and dead cells. These organoids were then fixed using PFA, rinsed using PBS, positioned in agarose microwells, and encapsulated in agarose blocks by pouring another layer of melted agarose solution on top. These agarose blocks could be sectioned into 50 μm slices and then used for immunostaining to analyze phenotypic changes in organoids (Figure 1B,C). By not having to process organoids one by one, we saved several hours on every batch of organoids we embedded. By arranging organoids in arrays, we not only saved time but also applied the same parameters in staining and imaging for reliable comparison among different groups. To accurately distinguish individual organoids, we trimmed the corner of the agarose block to label the orientation and arrangement of array organoids.

To 3D immunostain whole organoids in batches efficiently, we fabricated a microarray staining device for batch staining (Figure 1(A3)), which reduces the time needed during liquid changing and prevents the deformation of organoids in the process. Traditionally, to immunostain organoids, we had to immerse them in several types of solutions sequentially, with rinsing steps in between. The manual process of adding and removing solutions takes a lot of time and could easily cause deformation of the delicate structures of cerebral organoids. Organoids could be positioned in the batch staining device without falling through. After positioning organoids in the microwells of this device, we could simply lift the device and place it in another well with a different solution. These organoids were then embedded in agarose in arrays for 3D imaging by fMOST. To facilitate emersion via this approach, we attached the design files of the array mold and batch staining device as shown in Figure A1. The dimensions of the microwells could be modified to accommodate organoids of different sizes. Our embedding, staining, and imaging methods could greatly reduce the time needed for sample preparation and analysis. With this strategy, we were able to perform in situ phenotypic analysis of organoids with spatial information efficiently.

### 2.2. Cell Viability Assay and Live/Dead Imaging of Cerebral Organoids in Disease Modeling

To demonstrate the batch-parallel processing capacity of our strategy in organoid modeling, we constructed several types of disease models, including models of aging, neuroinflammation, and oxidative stress, and investigated their cellular and molecular characteristics using our strategy. We modeled oxidative stress using 1 mM hydrogen peroxide [22], neuroinflammation using 20 μg/mL lipopolysaccharides [23], aging using 60 mg/mL D-galactose [24], and aging therapy using 50 ng/mL neural growth factor (NGF) as a treatment [25]. To overcome the diffusion limitations, we selected the basal drug concentration from the 2D culture modeling and increased it by 2~3 times compared to the 2D culture modeling.

After drug treatments, we investigated the effects of these treatments on the size and cell viability of cerebral organoids. We assessed the size changes of organoids, calculated cell viability using cell counting kit-8 (CCK-8), and imaged the distribution of live/dead cells in whole organoids using a fluorescent microscope. We took photos of organoids under a light microscope to investigate whether there were any changes in the diameters of organoids after drug treatments (Figure 2A). We found that in the aging models and aging therapy models, organoids increased in diameter (*p* < 0.01, *p* < 0.05, respectively), and there was no change in diameter in models of neuroinflammation and oxidative stress (*p* > 0.05) (Figure 2B). By using CCK-8, we found that cell viability in models of oxidative stress and neuroinflammation decreased (*p* < 0.001), while the aging model and aging therapy model had the same cell viability compared to the control group (*p* > 0.05) (Figure 2C). We performed live/dead staining of whole organoids and imaged them on an agarose support, which allowed us to localize live and dead cells in each organoid in an array (Figure 2D,E). Organoids from all treatment groups were transferred from 96-well plates to microwells in agarose supports for live/dead imaging. Such a parallel imaging setup of multiple organoids saved time and facilitated easy comparison among different treatment groups. We performed these three assays on each of the organoids and recorded their data, respectively. We were able to trace all the data points obtained from each organoid and correlate the results in the analysis. The information flow was traceable, and the results from the live/dead assay and the CCK-8 assay could mutually verify the reliability of both tests.

### 2.3. In Situ Phenotypic Analysis in Sectioned Agarose Arrays of Cerebral Organoids in Aging Models

To analyze changes in the distribution of biomarkers at the single-cell level among treatment groups, we sectioned embedded organoid arrays into 50 μm thick slices for immunostaining and imaging. Here, we investigated changes in aging hallmarks in the aging model and the aging therapy model in particular. The phenotypes we analyzed included a neuronal marker (Microtubule-associated protein 2, or MAP2; Figure 3A), cytokines (interleukin-6, or IL-6; Figure 3B), glial fibrillary acidic protein (GFAP; Figure 3C), a cellular proliferation marker (Ki67; Figure 3C), tumor suppressor genes (p16 and p53), apoptosis (TUNEL assay; Figure 4), and a neural precursor marker (SOX2; Figure 4).

We arranged organoids from different groups in the same array so we could quantitatively analyze the expression of each marker across treatment groups (Figure 3A–C). The expression of MAP2 in the center of organoids in the aging model decreased compared to the control group and recovered slightly in the therapy group (Figure 3D–F), but the change in total expression was not statistically significant (*p* > 0.05). We quantitatively analyzed the expression of all the biomarkers we assessed and found that p16 was the only marker significantly increased in expression in the therapy group (*p* < 0.05, Figure 3G). By using this strategy, we were able to localize and quantitatively analyze the expression of multiple aging hallmarks in organoid arrays. In addition, we could track the data points and treatments of each organoid in the array when analyzing the results and comparing them among different treatment groups.

We also analyzed the changes in the stratification of cerebral organoids after treatments. In all groups, apoptotic cells were mostly positioned within organoids (inside the outer surface), as shown by the TUNEL assay, where cells had less access to oxygen and nutrients (Figure 4A). A higher density of neural stem cells was located close to the surface of organoids, as shown by SOX2 (Figure 4A), while a smaller population of neural stem cells was situated within the interior of the organoids (Figure 4C,E,G). This stratification is shown more clearly in the detailed graphs of the perimeters (Figure 4B,D,F) and the centers of organoids (Figure 4C,E,G). There was no colocalization between the two markers, showing that neural stem cells were not apoptotic.

Next, we analyzed the distribution of aging hallmarks p16 and IL-6 in the aging model and the aging therapy model. p16 was expressed in both the soma and axons of neurons and was localized more densely near the perimeter than in the center of organoids (Figure 4H). In the aging model, there was no change in the amount of p16 expressed, but there was increased p16 expression in the aging therapy model. The expression patterns of IL-6 changed after treatments (Figure 4I). IL-6 was only expressed in the somas in the control group, but both the somas and axons of neurons in the aging model and the aging therapy model expressed IL-6. The amount of IL-6 expressed increased slightly in the aging model and decreased slightly in the aging therapy model, but this was not statistically significant. We were able to show clear stratification in cerebral organoids by showing the spatial distribution and colocalization of different biomarkers at single cell levels. We were also able to show changes in expression patterns with high resolution in subcellular structures in disease models using this strategy.

### 2.4. Three-Dimensional In Situ Phenotypic Analysis Showed Changes at Single-Cell Levels and Subcellular Levels

To investigate the changes in cell morphology and spatial distribution of biomarkers in 3D in disease models, we performed 3D batch immunostaining and 3D array imaging using fMOST. First, to demonstrate 3D batch immunostaining, we 3D-immunostained organoids with GFAP for 3D imaging using fMOST with a voxel resolution of 0.32 μm × 0.32 μm × 2 μm. Maximum projection was performed over 25 raw images to create images of a 50 μm thick optical section of organoids (Figure 5A). We were able to observe that GFAP^+^ cells, one of the main cell types in cerebral organoids, were distributed throughout the entire organoids (Figure 5A). This result, on the other hand, showed that we were able to 3D immunostain the entire organoids, as we could observe GFAP^+^ cells at the center of those organoids. Then, to demonstrate 3D array imaging, we embedded organoids that had undergone different drug treatments and staining in agarose arrays (Figure 5B). The corners of the agarose block were trimmed to ensure accurate tracking of the orientation of the agarose block and the arrangement of embedded organoids. After batch 3D imaging, we were able to analyze the distribution of TUNEL, GFAP, and IL-6 in three treatment groups at the same time (Figure 5C). We found that the expression patterns of IL-6 were different in each group (Figure 5D). In the control group, IL-6 was mainly found in the somas only, but in the aging model, IL-6 was also observed on the axons. In the aging therapy model, the number of neurons that expressed IL-6 in both somas and axons was lower than that in the aging model but higher than that in the control group. This result from 3D analysis was consistent with that from sectioned organoid arrays (Figure 4I). In addition, we created 3D-rendered images showing the spatial localization of neurons, of which both the somas and axons were IL-6 positive. The numbers of such neurons in one cerebral organoid were 61, 216, and 156 in the control group, aging model, and aging therapy model, respectively (Figure 5E). We evaluated the spatial distribution of biomarkers in agarose sections and in 3D and were able to verify the results using the two approaches. We were able to visualize the expression patterns of biomarkers in subcellular structures like somas and axons in 3D for phenotypic analysis of organoids using this strategy. By combining 3D imaging and array embedding, we saved time and effort in organoid analysis by batch processing and were able to compare differences among treatment groups and localize affected biological structures.

## 3. Discussion

We designed an array mold to produce agarose supports with microwells for efficient batch staining and imaging. To illustrate the applications of array embedding and imaging strategies in disease models, we analyzed the changes in cell viability and expression of aging markers in organoid models in agarose sections and in 3D. The same imaging parameters were used to obtain images of every organoid in an array at once, which facilitated quantitative analysis and comparison among groups and saved a lot of time. The experimental results demonstrated that this strategy was suitable for evaluating disease models using organoids.

Batch phenotypic analysis of organoids has been a hotspot in the field of organoid research. These developments have led to increased efforts in high-throughput screening of organoids, including high-content imaging analysis and single-cell RNA sequencing technology, which was too expensive for use in common research laboratories [26,27]. There was no cheap and fast method for biomarker staining and imaging of cerebral organoids in batches [28]. Investigation of disease models using organoids has provided increasing needs for high-throughput drug screening methods.

Previous studies have shown that embedding in agarose did not disrupt imaging of fluorescent markers, and thus array embedding of organoids is a cheaper alternative for in situ phenotypic analyses in organoids [29,30]. Agarose array embedding has increased efficiency, leading to reduced time, effort, and research budgets. Our design of the batch staining device also aided in saving us more time while helping maintain the integrity of the delicate structures of cerebral organoids. With the help of this strategy, researchers could conduct drug screening tests with more treatment groups using more organoids in less time. Our study has provided a strategy that allows batch phenotypic analysis of organoids in disease models, including models of oxidative stress, neuroinflammation, aging, and aging therapy.

The comparison of expression patterns of biomarkers among different treatment groups in the same array would be more consistent and reliable because all the samples were embedded and imaged in parallel with exactly the same parameters. The capacity of this strategy to perform parallel experiments with easily comparable results could help save time and increase the reliability of the results. In addition, we have records of all the history of each organoid and could analyze their phenotypic changes after their specific treatments in such parallel experimental settings.

While there are other methods to embed organoids in arrays, they are more labor-intensive than our design of the resin mold and agarose support. Organoids or spheroids could be embedded in paraffin or cryomolds, followed by arrangement in arrays using a microarrayer [28]. These methods could work for organoids and spheroids but require considerably more time, and the processes are more complex. Other studies using agarose molds for array embedding did not provide a complete pipeline for organoid batch processing, which includes batch staining and 3D imaging [19,21]. These protocols also only demonstrated agarose array embedding for small spheroids of several hundred micrometers. The size of organoids often leads to additional challenges in sample processing and whole-volume imaging, and some cerebral organoids have already reached several millimeters in diameter [12,13]. With our strategy, we were able to embed and image complex cerebral organoids greater than one millimeter in diameter.

Our current strategy could only allow simultaneous analysis of several organoids in an array. Our molds were only 3 × 3 or 3 × 5 microwells, allowing at most 15 organoids to be imaged and analyzed at the same time. This could be easily scalable by designing molds with more microwells. However, in disease modeling and drug screening, there are usually many types of experiments to be performed to evaluate the changes in phenotypes in organoids after treatments, and immunostaining is only one of them. We would not need to immunostain hundreds of organoids. Our strategy provided a fast and easy method for sample inspection. We could evaluate the spatial distribution of certain biomarkers quickly and efficiently by analyzing a few organoids from each treatment group using this strategy. In the future, we plan to further improve this methodology for more and larger organoids phenotypic analysis. The incorporation of these larger organoids holds the promise of heightened efficacy in illustrating the prowess of our 3D imaging methodology. We also hope to develop a more comprehensive strategy combined with computer-based automated image analysis capable of processing image data obtained from hundreds of organoids in a short time without much human intervention. We would like to design a new device in which we can culture, drug-treat, fix, and stain organoids. Such a device would bridge the current gap between organoid culturing and whole-mount staining.

## 4. Conclusions

In conclusion, we developed a comprehensive strategy for staining, embedding, and imaging organoids for phenotypic analysis of disease models using both agarose sections and 3D imaging. This strategy is easy to implement in most laboratories and does not require extensive time and effort. By performing parallel analyses of different treatment groups, researchers could easily compare groups using more reliable experimental results. This approach could also be applied to other types of organoids other than cerebral organoids and be valuable in applications in drug screening, disease modeling, and organs on a chip.

## 5. Materials and Methods

### 5.1. Drug Treatment

Organoids (Cat # HopCell-3D-C), with diameters ranging from 1 to 1.5 mm, were purchased from Hopstem Biotechnology Inc. (Hangzhou, China), and the generation of these organoids followed a published protocol [31]. Organoids were cultured in 5% CO_2_ incubators at 37 °C in 96-well plates made by NEST technology. The growth medium used was 3D cerebral organoid medium (Cat # HopCell-3DM-100A) with a 3D cerebral organoid medium supplement (Cat # HopCell-3DM-100B). Half of the medium was replaced every three days.

We added 200 μL of media containing 1 mM hydrogen peroxide, 20 μg/mL LPS, 60 mg/mL D-galactose, and 60mg/mL D-galactose + 50 ng/mL NGF to wells in 96-well U-bottom plates containing cerebral organoids to create models of oxidative stress, neuroinflammation, aging, and aging therapy, respectively. The organoids had been cultured for 220 days at the time of drug treatment. The treatments lasted 48 h and there were at least six organoids in the experimental group.

### 5.2. Cell Viability Assay and Live Imaging

To determine changes in cell viability in organoids after treatments, cerebral organoids were incubated in a 10% CCK-8 solution diluted using 3D cerebral organoid medium for 2 h, and the absorbance at 450 nm was measured using a microplate reader for the calculation of cell viability. To obtain the spatial localization of live and dead cells in organoids, cerebral organoids were incubated in 2 μM Calcein-AM and EthD-1 (L3224, Thermo Fisher, Waltham, MA, USA) for 30 min at 37 °C in an incubator. These organoids were placed in agarose supports for array imaging under a fluorescent stereo zoom microscope (Zeiss Axio Zoom.V16, Carl Zeiss, Oberkochen, Germany). The images taken were at 7× and 40×.

### 5.3. Designing and Rapid Prototyping of the Array Mold

The array mold was designed using SolidWorks (Premium 2017) and 3D-printed using light-sensitive resin. The 3D printer sequentially solidifies liquid photosensitive resin layer by layer to complete a section painting operation. Multiple interfaces were overlapped to complete 3D solid printing with an accuracy of 0.1 mm.

### 5.4. Array Agarose Embedding

Agarose (5% wt) was placed in an Erlenmeyer flask containing 0.01 M PBS solution and 10 mM sodium periodate. The flask was covered with aluminum foil to keep the liquid dark, and the flask was placed on a magnetic stirrer for two to three hours to oxidize the agarose. The agarose suspension was filtered using a Büchner funnel connected to a vacuum. The agarose was washed with 0.01 M PBS several times to remove all the sodium periodate. The oxidized agarose was collected in another Erlenmeyer flask, and 50% wt glycerol and 50% 0.01 M PBS solution were added to make a 5% wt agarose suspension. The agarose suspension was heated in a microwave oven until all the agarose was melted, and the flask was placed in a water bath at 55 °C.

The resin mold was placed in a silica gel mold, and 3 mL of melted agarose solution was added on top of the resin mold. The agarose was let sit at room temperature for ten minutes to solidify and become agarose support with microwells. The microwells on the agarose support were rinsed with 0.01 M PBS, and cerebral organoids were placed in the microwells. Then, 3 mL of melted agarose solution was poured on top of the organoids to encapsulate them, and the agarose block was kept in the water bath for at least twenty minutes for the agarose and organoids to crosslink. The agarose block was placed at room temperature for another ten minutes and was ready for sectioning and imaging.

### 5.5. Cell Apoptosis Detection Using the TUNEL Assay

Organoids or organoid array sections were washed with PBS three times for five minutes, and they were incubated in 50 μL of TUNEL reagents at 37 °C for 60 min, followed by rinsing with PBS. After staining, the organoid array sections were mounted on glass slides with 50% glycerol under coverslips and imaged using a confocal microscope. Whole organoids were imaged using the fMOST system.

### 5.6. Immunostaining

The agarose blocks containing cerebral organoids were sectioned into 50 μm slices using a vibrating blade microtome (Leica VT 1200S, Leica Microsystems, Wetzlar, Germany). The cutting speed was 1 mm/s, and the amplitude was 1 mm. The sections were rinsed using PBS three times for five minutes and then placed in a blocking buffer of 5% BSA and 0.3% TritonX-100 PBS solution for 2 h at 37 °C to prevent nonspecific binding. After blocking, sections were stained with primary antibodies diluted in the blocking buffer at 4 °C overnight. The next day, the sections were washed with PBS three times for five minutes. Then, the sections were stained with secondary antibodies diluted in PBS for two hours at 37 °C. The sections were washed with PBS three times for five minutes and then placed in a 1 μg/mL DAPI solution to stain cell nuclei for ten minutes. The sections were washed with PBS three times for five minutes. Finally, the sections were placed in PBS solutions and mounted to glass slides with 50% glycerol. The mounted sections were imaged using confocal microscopes.

### 5.7. Three-Dimensional Immunostaining

We immunostained the intact cerebral organoids with a diameter of 1.5 mm using the iDISCO protocol [32]. All samples were fixed in a 4% paraformaldehyde solution for 2 h and then washed in PBS 1 h. A graded methanol/H_2_O series (20%, 40%, 60%, 80%, 100%, and a second 100%) was used to dehydrate samples for 30 min each at 4 °C. Samples were bleached in chilled 5% H_2_O_2_/20% DMSO/methanol overnight at 4 °C. Next, another graded methanol/H_2_O series (100%, 80%, 60%, 40%, and 20%) was used to rehydrate samples for 30 min each at 4 °C. Then, samples were incubated in PBS/0.5% Triton X-100 (PTX) for 12h at 4 °C. To perform immunostaining, samples were treated in PTX/20% DMSO/0.3 M glycine at 4 °C for 12 h, then blocked in PTX/10% DMSO/6% BSA at 4 °C for 12 h. Samples were washed in PBS/0.2% Tween-20 (PTw) at 37 °C overnight. Samples were incubated with the primary antibody in PTw/5% DMSO/3% BSA at 4 °C for four days and washed in PTw for 1 h, five times. Samples were incubated with the secondary antibody in PTw/3% BSA at 37 °C for two days. Finally, samples were washed in PTw for 1h five times. The concentrations of primary and secondary antibodies used are shown in Table 1.

### 5.8. Microscopic Imaging

To preliminarily judge the effect of drugs on organoids, the stained organoids in arrays were imaged by a commercial confocal microscope (LSM710, Carl Zeiss, Oberkochen, Germany) with 10× and 20× objective lenses. The fMOST system [33,34] was used to obtain dual-wavelength and high-resolution 3D data on organoids. During imaging, the sample was immersed in a water bath. The imaging system performed data acquisition at a voxel resolution of 0.32 μm × 0.32 μm × 2 μm. Mosaic-by-mosaic scans of the sample surface were performed by moving the 3D precision stage. After one layer of images was acquired, the imaged layer was excised. Then, the imaging system continued to acquire images of the next layer. The cycle repeats until the entire organoid is imaged.

### 5.9. Visualization and Reconstruction

We converted microscopic images into TIFF format and built z-stacks using ImageJ software (NIH, Bethesda, MD, USA). We processed the data using Imaris (Bitplane, Zurich, Switzerland v9.0) to visualize 3D reconstruction. For quantitative statistical analysis, we use ImageJ software to import the desired site cut into Imaris software to form a 3D data block and perform cell counting using the Spots function and 3D visualization.

### 5.10. Statistical Analysis

Data were analyzed in GraphPad Prism version 8. All data were presented as mean ± SD, and ANOVA analysis was used to compare more than two groups of data points. In this study, *p* < 0.05 was considered significant (* *p* < 0.05, ** *p* < 0.01, *** *p* < 0.001).

## Figures and Tables

**Figure 1 ijms-24-13903-f001:**
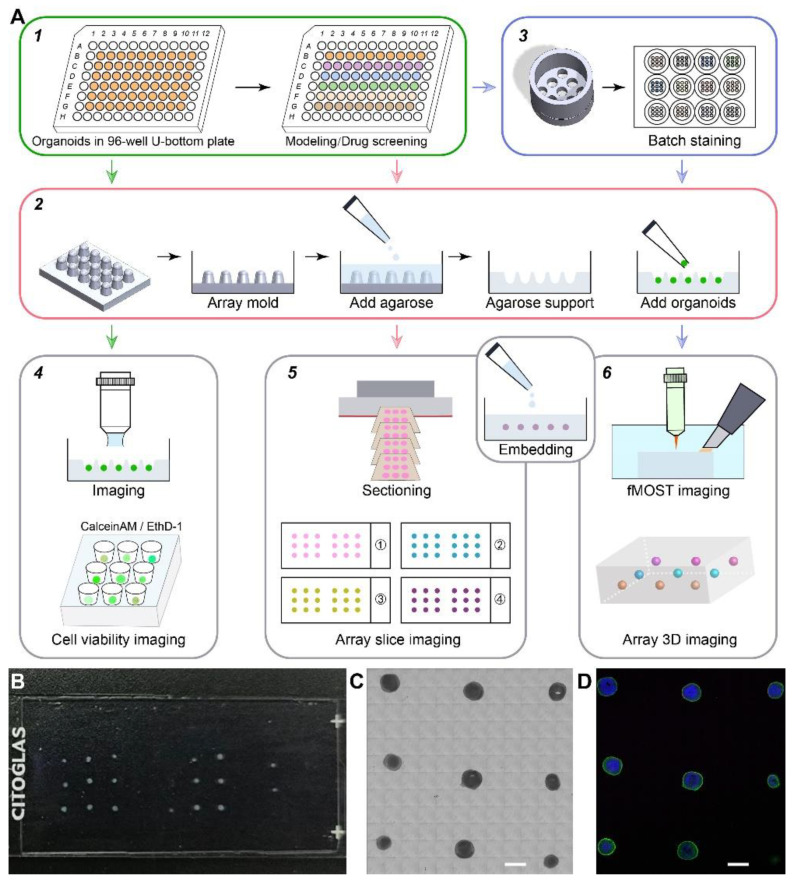
Scheme for disease modeling, array embedding, batch immunostaining, and imaging of cerebral organoids. (**A**). Strategies for disease modeling and phenotypic analysis using cerebral organoids. (1). Modeling and drug screening in 96-well plates. (2). Preparation of the agarose support. (3). 3D batch staining. (4). Live/dead cell imaging in agarose microwells. (5). Fluorescent imaging of multi-stained organoid arrays embedded in agarose. (6). 3D whole-volume imaging of organoid arrays. (**B**). Sectioned agarose slices with organoid arrays on a glass slide. (**C**). View of organoid arrays under a light microscope. Scale bar: 1 mm. (**D**). Immunostained organoid arrays under a fluorescent microscope. Scale bar: 1 mm.

**Figure 2 ijms-24-13903-f002:**
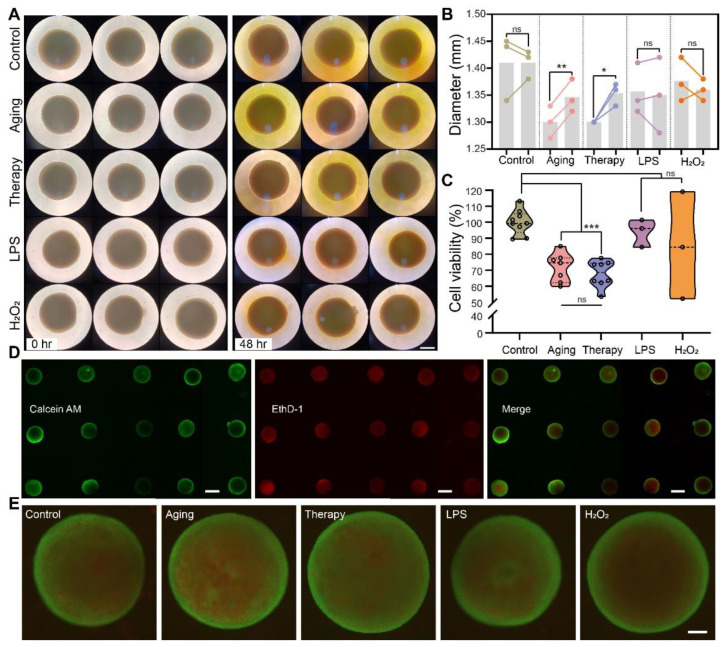
Application of array imaging in cerebral organoid disease models. (**A**) Size comparison of cerebral organoids in different treatment groups. The group on the left showed organoids before drug treatments, and the group on the right showed organoids 48 h after drug treatments. Scale bar: 500 μm. (**B**) Statistical analysis of the changes in diameters of cerebral organoids in different treatment groups using ANOVA. The first bars in each group show the diameters of organoids before drug treatments, and the second bars of each group show that after 48 h drug treatments, * *p* < 0.05, ** *p* < 0.01; ns, no significance. (**C**) Statistical analysis of cell viability using CCK-8 in different treatment groups using ANOVA, *** *p* < 0.001. Each point in the graph represents the cell viability of one organoid 48 h after treatments. (**D**) Live/dead stained organoids in arrays. Scale bar: 1 mm. (**E**) Detailed images of live/dead stained organoids from different treatment groups. Scale bar: 200 μm.

**Figure 3 ijms-24-13903-f003:**
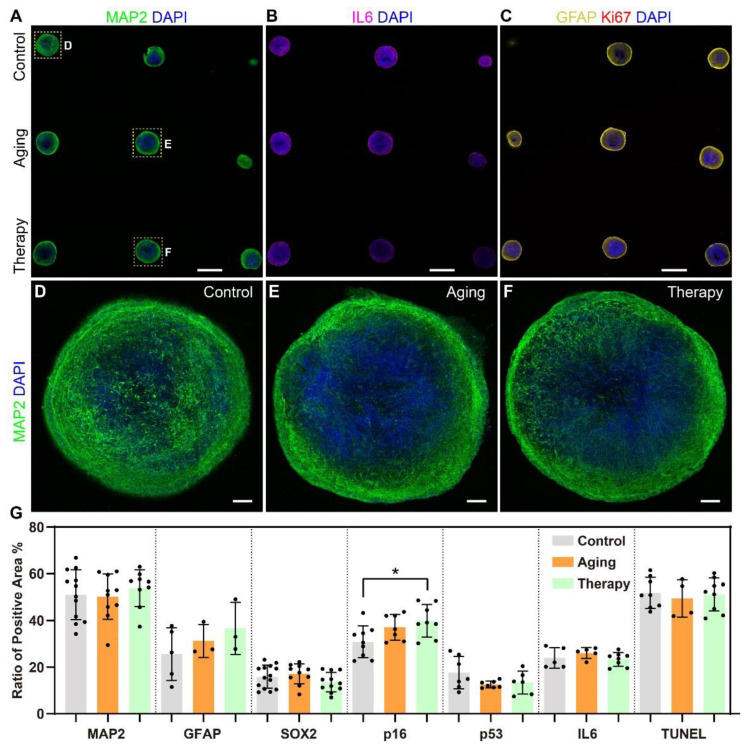
Sectioned organoid arrays for analyzing aging hallmarks in aging models using cerebral organoids. (**A**–**C**) Sectioned organoids were stained with MAP2, IL-6, GFAP, and Ki67, while DAPI stained the cell nucleus. Scale bar: 1 mm. (**D**–**F**) Detailed images of organoids from different treatment groups showing the changes in expression of MAP2 and DAPI. Scale bar: 100 μm. (**G**) Statistical analysis of changes in marker expression using ANOVA, * *p* < 0.05. Each point in the graph represents the ratio of positive area in one organoid after different treatments.

**Figure 4 ijms-24-13903-f004:**
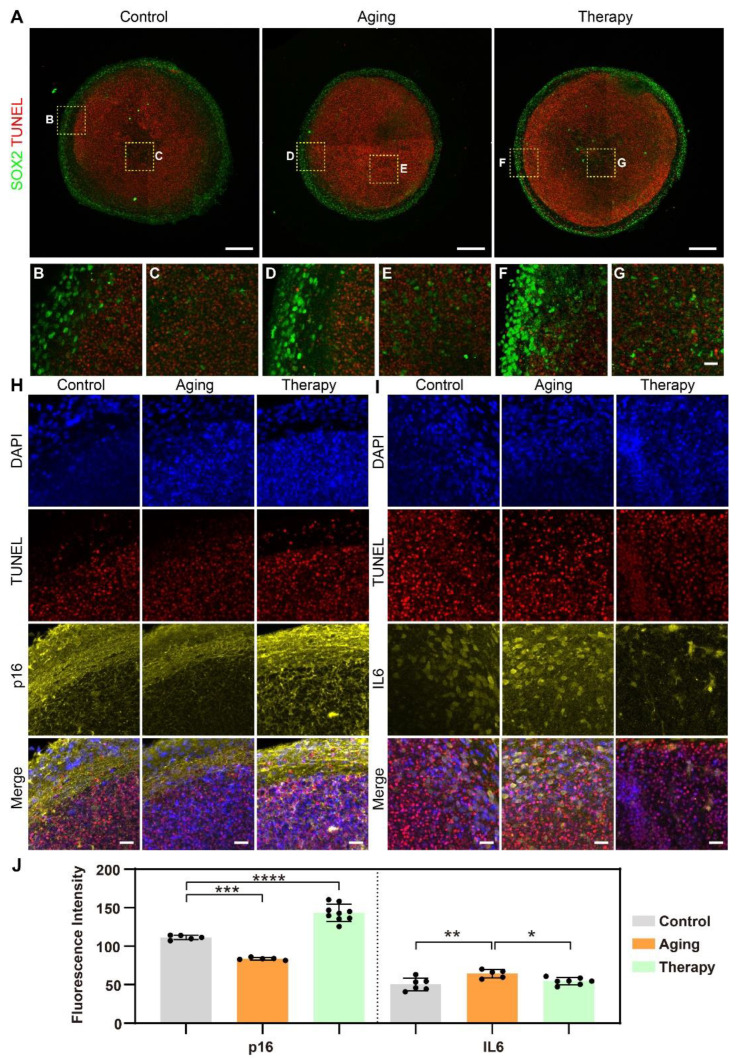
Distribution analysis of apoptosis, neural stem cells, and aging hallmarks in aging models using cerebral organoids. (**A**) Cerebral organoids stained with the TUNEL assay and SOX2. Scale bar: 200 μm. (**B**–**G**) Detailed images of organoids as denoted in (**A**). Scale bar: 25 μm. (**H**) Cerebral organoids stained with DAPI, TUNEL assay, and p16. Scale bar: 25 μm. (**I**). Cerebral organoids stained with DAPI, TUNEL assay, and IL-6. Scale bar: 25 μm. (**J**) Statistical analysis of expression of aging hallmarks in different treatment groups using ANOVA, * *p* < 0.05, ** *p* < 0.01, *** *p* < 0.001, **** *p* < 0.0001. Each point in the graph represented the fluorescence intensity in one organoid after different treatments.

**Figure 5 ijms-24-13903-f005:**
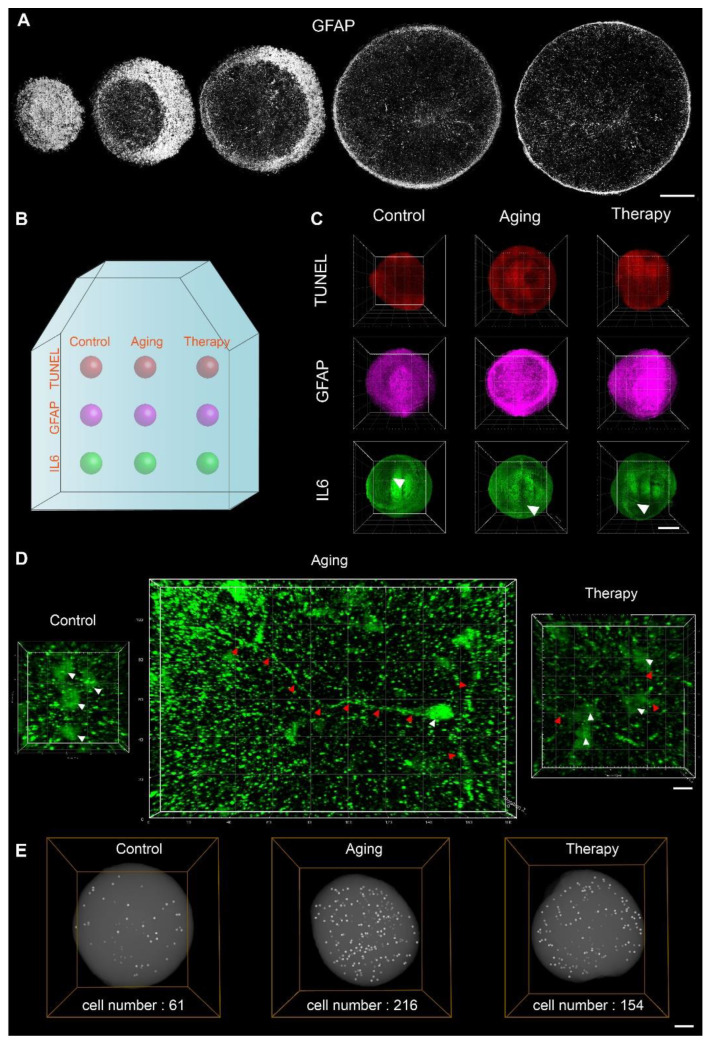
Three-dimensional imaging of array-embedded cerebral organoids for analysis of aging models. (**A**) Distribution of GFAP^+^ cells in organoids on a series of 50 μm thick slices that were created by maximum projection using the fMOST dataset. Scale bar: 200 μm. (**B**) Diagram showing an agarose block with organoids embedded in an array. Corners were trimmed to mark the orientation of the agarose block and the arrangement of individual organoids. (**C**) Three-dimensionally rendered images of immunostained cerebral organoids in an array using fMOST. A 3D view of organoids provided a comprehensive view that extended beyond the organoid surfaces. The area of white arrows in IL-6-stained organoids is enlarged in Figure 5D for a clearer view. Scale bar: 200 μm. (**D**) Change in distribution patterns of IL-6 in aging models and aging therapy models in 3D rendered images of organoids. White arrows denoted somas, and red arrows denoted axons. Scale bar: 10 μm. (**E**) Three-dimensionally rendered graphs showing the spatial localization of neurons, of which both the somas and axons were IL-6 positive. Scale bar: 100 μm.

**Table 1 ijms-24-13903-t001:** Antibodies used in the study.

Antibody	Dilution for Sections	Dilution for 3D Immunostaining	Cat. No.	Company
Anti-p16, mouse	1:100	-	Sc-1661	Santa Cruz Bio, Dallas, Texas, USA
Anti-p53, mouse	1:100	-	Sc-126	Santa Cruz Bio, Dallas, Texas, USA
Anti-IL6, mouse	1:100	1:10	Sc-57315	Santa Cruz Bio, Dallas, Texas, USA
Anti-Ki67, mouse	1:20	1:10	550609	BD Pharmingen, San Diego, CA, USA
Anti-SOX2, rabbit	1:500	1:50	ab97959	Abcam, Cambridge, UK
Anti-MAP2, rabbit	1:500	1:50	AB5622	Millipore, Darmstadt, Germany
Anti-GFAP, chicken	1:500	1:50	C9205	Millipore, Darmstadt, Germany
Alexa Fluor 488 donkey anti-chicken lgG (H + L)	1:1000	1:100	A78948	Invitrogen, Waltham, MA, USA
Alexa Fluor 488 donkey anti-rabbit lgG (H + L)	1:1000	1:100	A21206	Invitrogen, Waltham, MA, USA
Alexa Fluor 594 donkey anti-mouse lgG (H + L)	1:1000	1:100	A21203	Invitrogen, Waltham, MA, USA

## Data Availability

All data related to this study are contained within the manuscript. Data can be obtained from the corresponding authors on request.

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
