# Peer review of "Cerebral Organoid Arrays for Batch Phenotypic Analysis in Sections and Three Dimensions"

_ijms, 2023, doi:10.3390/ijms241813903_

Round 1

Reviewer 1 Report

In this manuscript, Chen et al. aimed to improve the analysis of brain organoids towards high-throughput using agarose microwells. As a proof of principle, the authors used an aging model in organoids. For me the advantage of this strategy/method is not convincing. The usage of agarose molds/microwells allow to analyze 9-15 organoids in parallel. This is not an advantage at all. It is pretty standard in the field to embed several organoids in one OCT block, section this block and image the organoids from this block. For 3D imaging of brain organoids agarose embedding is also typically used and the usage of agarose microwells is not a major improvement in comparison to previous methods. Moreover, the authors claim that their method would allow high-throughput analysis but how can the parallel analysis of 9-15 organoids be called high-throughput? Especially, as the bottleneck is currently the image analysis. In summary, the authors only present an incremental improvement in the analysis of brain organoids, which does not justify a publication in International Journal of Molecular Sciences. I, therefore, recommend rejecting this manuscript.

Besides these major concerns there are several issues which need to be addressed:

1) What is/are the age(s) of the organoids used in this study. It is whether mentioned in the text nor in the figure legends. The authors claim that their approach in comparison to previous studies allows the analysis of large organoids. However, the organoids presented are just around 1mm in diameter and brain organoids are well known to reach up to 5 mm. So, I don’t see the major improvement to previous methods. Moreover, their brain organoids are missing the typical ventricle-like structures or neural rosettes (as they call them). This is typical for either failed organoids or very early organoids maybe even at embryoid body stage. The authors need to clearly state the age/stage of organoid in the text and figure legends.

2) In Figure 2B, it is unclear what the authors present here. They compare within groups but what is for example the difference between control first bar vs. control second bar? Are these different time points, before/after treatment? The authors need to explain this either in the text or in the figure legend. Moreover, Figure 2B is not mentioned in the text at all.

3) In the same line, all figure legends miss important information, e.g. organoid stage, what the data point stand for (see Figure 2C, are these organoids?), statistical tests, etc.

4) In line 167/8 The authors mention that MAP2 is a marker for neural subtypes. However, MAP2 is a marker for neuronal differentiation, as it is mainly found in neurites.

5) Figure 4A shows that almost everything except a thin layer of cells is dead in the presented organoids. This again does not speak for the quality of the organoids used in this study!

6) In Figure 5C, the authors present 3D imaging of the organoids. However, the authors only show images of the surface of the organoids. To proof that the staining and imaging worked, the authors should show an optical section through the 3D imaged organoid to prove that the staining really penetrated the organoid and is not just a surface staining.

7) The authors claim that the batch staining device is a strong improvement, while this statement is questionable (see above), it would be nice if the authors would provide blueprints and/or designs/files for the 3D printer so that other researchers could benefit and use the “advantage” of this approach.

Author Response

Reviewers 1

In this manuscript, Chen et al. aimed to improve the analysis of brain organoids towards high-throughput using agarose microwells. As a proof of principle, the authors used an aging model in organoids. For me the advantage of this strategy/method is not convincing. The usage of agarose molds/microwells allow to analyze 9-15 organoids in parallel. This is not an advantage at all. It is pretty standard in the field to embed several organoids in one OCT block, section this block and image the organoids from this block. For 3D imaging of brain organoids agarose embedding is also typically used and the usage of agarose microwells is not a major improvement in comparison to previous methods. Moreover, the authors claim that their method would allow high-throughput analysis but how can the parallel analysis of 9-15 organoids be called high-throughput? Especially, as the bottleneck is currently the image analysis. In summary, the authors only present an incremental improvement in the analysis of brain organoids, which does not justify a publication in International Journal of Molecular Sciences. I, therefore, recommend rejecting this manuscript.

Response: Thanks for your comments. This article aims to establish the viability of analyzing multiple organoids through this modified methodology. The proposed method exhibits a capacity for seamless scalability, thereby facilitating the analysis of an increased number of organoids. This scalability is achieved through the strategic design of expanded agarose molds. In the immediate future, our research agenda involves the implementation of agarose molds featuring amplified rows and columns, thereby accommodating more and higher volume of organoids for the purposes of comprehensive and efficient analysis. To accurately present this study, we revised the title without of the term "high-throughput."

1) What is/are the age(s) of the organoids used in this study. It is whether mentioned in the text nor in the figure legends. The authors claim that their approach in comparison to previous studies allows the analysis of large organoids. However, the organoids presented are just around 1mm in diameter and brain organoids are well known to reach up to 5 mm. So, I don’t see the major improvement to previous methods. Moreover, their brain organoids are missing the typical ventricle-like structures or neural rosettes (as they call them). This is typical for either failed organoids or very early organoids maybe even at embryoid body stage. The authors need to clearly state the age/stage of organoid in the text and figure legends.

Response: Following this comment, we have incorporated the age of the organoids and the brief description of organoids culture within the Methods section (Lines 363-369 and 373-374), thereby enhancing the contextual accuracy of our study. The examined organoids in this study, characterized by diameters ranging from 1 mm to 1.5 mm, were effectively processed utilizing our devised methodology. Our method's efficacy should be able to extend to larger organoids without a lot of modifications. The microwells within our agarose molds and batch staining device could be enlarged to accommodate larger organoids for embedding purposes. In the future work, we plan further improve this methodology for more and larger organoids phenotypic analysis. The incorporation of these larger organoids holds the promise of heightened efficacy in illustrating the prowess of our 3D imaging methodology. We added this discussion in lines 343-346.

In this study, we consistently utilized the commercial organoids as presentation in our previous publication in 2022, and we elucidated neural rosette structures utilizing PI staining for labeling cytoarchitecture in 3D images (Fig. 3d, Ma et al., 2022). We also added this information in the revised manuscript (Lines 49-53).

2) In Figure 2B, it is unclear what the authors present here. They compare within groups but what is for example the difference between control first bar vs. control second bar? Are these different time points, before/after treatment? The authors need to explain this either in the text or in the figure legend. Moreover, Figure 2B is not mentioned in the text at all.

Response: In Figure 2B, we compare and investigate whether several types of disease models changes in diameters of the organoids before and after 48h-drug treatments. We added this detailed description and claimed this point in main text (Lines 155-160) and figure legend (Lines 175-182).

3) In the same line, all figure legends miss important information, e.g. organoid stage, what the data point stand for (see Figure 2C, are these organoids?), statistical tests, etc.

Response: Following your suggestion, we have improved all figure legends and clearly stated the data point indication and statistical tests method (as shown in figure legends of Figure 2/3/4), which were also mentioned in Statistical analysis section (Lines 462-465). All organoids were cultured for 220 days at the time of drug treatments (Lines 373-374).

4) In line 167/8 The authors mention that MAP2 is a marker for neural subtypes. However, MAP2 is a marker for neuronal differentiation, as it is mainly found in neurites.

Response: Microtubule-associated protein 2 (MAP2) is a neuronal differentiation marker (Soltani et al. Am J Pathol, 2005), and meanwhile it is highly expressed in neuronal somas and dendrites and specific enough to serve as robust mature neuronal marker (Lancaster et al. Nat Biotechnol, 2017; Renner et al. eLife, 2020). We have modified this description for accurate presentation (Lines 189-190).

5) Figure 4A shows that almost everything except a thin layer of cells is dead in the presented organoids. This again does not speak for the quality of the organoids used in this study!

Response: Fig. 4A does not pertain to a live/dead staining experiment. Instead, it portrays the outcomes of a TUNEL assay, highlighting apoptotic cells through a red signal, juxtaposed with SOX2 expression, denoted by green fluorescence, thereby delineating stem cells. It is known in the field that apoptotic cells are located inside organoids because of lack oxygen, etc. We did not show a cell nucleus staining in the same image so it cannot be concluded from this image that most of the cells are dead.

As shown by the image, the concentration of neural stem cells proximate to the organoids' outer periphery. Cells in this area have more access to oxygen and nutrients so it is not surprising that these cells are more actively dividing. Nevertheless, as presented in Fig. 4C, E, and G, a notable population of neural stem cells is discernibly situated within the interior of the organoids, showing the quality of organoids. We have modified the text to better describe this figure (Lines 213-217).

6) In Figure 5C, the authors present 3D imaging of the organoids. However, the authors only show images of the surface of the organoids. To proof that the staining and imaging worked, the authors should show an optical section through the 3D imaged organoid to prove that the staining really penetrated the organoid and is not just a surface staining.

Response: Fig. 5C showed 3D rendered images of organoids, not images of the surface of the organoids. These rendered images present a 3D view of the organoids, providing the readers a comprehensive view that extends beyond the organoid surface. And, Fig. 5A showed an organoid 3D immunostained with GFAP and imaged using fMOST with a z-resolution of 2 μm. Then, maximum projection over 25 images were performed to create images of 50-μm-thick optical section of organoids. This graph showed that GFAP+ cells were distributed throughout the entire organoid and also demonstrated that the staining really penetrated the entire organoid. The area of organoids denoted by white arrows in Fig. 5C are magnified in Fig. 5D, showing the detailed 3D distribution of IL-6 in the organoids. We have revised the main text (Lines 248-255) and figure legend (Lines 277-286) for clearer presentation.

7) The authors claim that the batch staining device is a strong improvement, while this statement is questionable (see above), it would be nice if the authors would provide blueprints and/or designs/files for the 3D printer so that other researchers could benefit and use the “advantage” of this approach.

Response: Following your suggestion, we have attached the design files of the agarose embedding mold and batch staining device as Supplementary Fig. 1 as shown in Supplementary Figures section (Lines 480-484) and revised the text (Lines 135-137).

Reviewer 2 Report

The authors demonstrated a novel method to check the phenotypic changes of 3D cell models. The need for large-scale phenotypic analysis of 3D platforms is one of the limitations in the high throughput analysis field in the drug discovery field.

However, I have a few quires from the data.

1. In Figure 1, each agarose well should display a single organoid, but the methods section does not specify how to select individual organoids per well. This information should be included.

2. Authors have mentioned 2D insitu phenotypic analysis of organoids along with 3D. I didn’t see any 2D (monolayer) culture in the manuscript. If staining is done on cryosections, terminology must be changed throughout the manuscript. Otherwise, the terminology is misleading the information. Authors can use whole organoid staining for 3D and cryosections instead of 2D.

For more details about 2D vs. 3D cultures, please follow these articles,

A. https://doi.org/10.1042/EBC20200150

B.  https://doi.org/10.3390/ijms24031912

3. Cerebral organoids take a long time for maturation. Which stage of organoids were used for modeling? Even though the authors used commercially available organoids, the details must be included in the methods section.

4. How do authors choose concentrations for disease modeling and treatment? Concentrations picked from 2D cultures may not be suitable for 3D cultures due to diffusion limitations, so optimization is necessary.

Author Response

Reviewers 2

The authors demonstrated a novel method to check the phenotypic changes of 3D cell models. The need for large-scale phenotypic analysis of 3D platforms is one of the limitations in the high throughput analysis field in the drug discovery field.

However, I have a few quires from the data.

  1. In Figure 1, each agarose well should display a single organoid, but the methods section does not specify how to select individual organoids per well. This information should be included.

Response: During batch staining, we group organoids with the same treatments into the same well. Individual organoids are positioned within the wells of the batch staining device, enabling clear identification of their well origins within the 96-well plates. For organoid embedding in agarose, a corner of the agarose block is trimmed. This step ensures accurate tracking of the agarose block's orientation and the arrangement of embedded organoids, as demonstrated in Fig. 5B. These clarifications have been incorporated into the figure legend of Fig. 5B and the main text (Lines 114-115 and 255-260).

  1. Authors have mentioned 2D in situ phenotypic analysis of organoids along with 3D. I didn’t see any 2D (monolayer) culture in the manuscript. If staining is done on cryosections, terminology must be changed throughout the manuscript. Otherwise, the terminology is misleading the information. Authors can use whole organoid staining for 3D and cryosections instead of 2D.

For more details about 2D vs. 3D cultures, please follow these articles,

  1. https://doi.org/10.1042/EBC20200150
  2. https://doi.org/10.3390/ijms24031912

Response: Thanks for the comment, we revised the phrasing of our description wherever we previously used the term 2D to refer to tissue sections in title and main text (such as, Lines 18-21 and 184-185).

  1. Cerebral organoids take a long time for maturation. Which stage of organoids were used for modeling? Even though the authors used commercially available organoids, the details must be included in the methods section.

Response: Following this suggestion, we briefly describe the organoid cultures, and added the cultured days of organoids in the method section (Lines 363-369 and 373-374).

  1. How do authors choose concentrations for disease modeling and treatment? Concentrations picked from 2D cultures may not be suitable for 3D cultures due to diffusion limitations, so optimization is necessary.

Response: In this study, the basal concentrations for disease modeling were picked from 2D cultures, but we increased the drug concentration to overcome the diffusion limitations, which is 2~3 times compared with 2D cultures modeling. We added this selection criteria of these 3D disease modeling using different drugs (Lines 149-151).

Round 2

Reviewer 1 Report

The authors have addressed all my comments.

Reviewer 2 Report

Thank you for the update. The manuscript has become more informative now.

There is room for improvement in regard to scientific writing.